# Impact of Land Use/Land Cover and Landscape Pattern on Water Quality in Dianchi Lake Basin, Southwest of China

Zhuoya Zhang [1,2], Jiaxi Li [2], Zheneng Hu [3], Wanxiong Zhang [2], Hailong Ge [2] and Xiaona Li [1,2,*]

[1] College of Geography and Eco-Tourism, Southwest Forestry University, Kunming 650224, China
[2] Ecological Civilization Research Center of Southwest China, National Forestry and Grassland Administration, Southwest Forestry University, Kunming 650224, China
[3] School of Economics, Yunnan University, Kunming 650500, China
[*] Correspondence: xiaonali@swfu.edu.cn

**Abstract:** The water quality of a basin is pronouncedly affected by the surrounding types of land use. Analyzing the impact of LULC and landscape patterns on water quality is critical for identifying potential drivers. To further study how LUCC affects the water quality in a typical plateau lake basin, this study investigated the impacts of land-use types on water quality in the Dianchi Lake Basin in Southwest China. We analyzed changes in land-use types and the landscape pattern of the Dianchi basin, calculated the CWQI (Canadian Water Quality Index) value based on the water quality indexes (PH, total phosphorus (TP), total nitrogen (TN), chemical oxygen demand (COD), dissolved oxygen (DO), permanganate index ($COD_{Mn}$), five-day biochemical ox-ygen demand ($BOD_5$), ammonia nitrogen ($NH_3$-N), turbitidy, and chlorophyll-a (Chla)), used the RDA (Redundancy Analysis) and SMLR (Stepwise multiple linear regression) methods, the coupling degree, coupling coordination degree, and the geographical detector model to explore the relationship between water quality and changes in the land-use type. The results show that (1) changes in the land-use types were obvious: the majority of the land, which was originally forest land, became built land in 2020 and farmland in 1990 (except for the Dianchi water). Landscape pattern indexes indicated that almost all land-use types were first scattered, then gathered from 1990 to 2020. (2) Changes in the water quality of Dianchi Lake lagged behind the changes in land-use types, and the variation trends were similar to the landscape pattern variation trends. The CWQI value decreased in a nearly linear fashion from 1990 to 1998, exhibited a slight change from 1999 to 2013, and quickly increased from 2013. (3) Land-use types demonstrated a tight correlation with the Dianchi water quality, and LPI was the most dominant factor in both Caohai Lake and Waihai Lake. (4) There were different indexes affecting the coupling coordination degrees of Caohai Lake and Waihai Lake.

**Keywords:** water quality; landscape patterns; coupling coordination; geographical detector; Dianchi Lake Basin

## 1. Introduction

Water quality is important for ensuring the safety of drinking water, a healthy living environment, and the sustainable development of surrounding areas [1–3]. Studies have shown that hundreds of river systems worldwide are severely polluted and degraded by changes in land use/land cover (LULC) and associated anthropogenic activities [4–6]. LULC and landscape patterns, which are a combination of natural and human disturbances, profoundly affect regional hydrological processes and water quality [7,8]. The relationship between water quality and the surrounding land's landscape characteristics is often affected by temporal and spatial changes [5,9]. As the main carrier of non-point source pollution, land affects changes in water quality through various complex processes driven by rainfall and runoff. This impact is a comprehensive response at different scales by multiple landscape structures [10–12], including LULC (i.e., different land-use types) and

landscape pattern (i.e., the spatial distribution and distribution of landscape patches) [12]. Therefore, discussing the impact of LULC and landscape patterns on water quality is critical for identifying potential drivers [13].

Studies have shown that changes in land use/cover types significantly change the water pollution status, and the resulting changes in the landscape pattern index are also significantly correlated with water quality [14]. LULC refers to the proportion of different land-use types [15]. Landscape patterns refer to the spatial structure and distribution characteristics of all elements in a landscape. They are usually characterized and quantified with landscape metrics. Landscape metrics are usually used to describe the landscape ecosystem, format, and trends in the landscape change for analyzing the interactions between land use and anthropogenic activities in watersheds [16,17]. Some researchers analyzed the impacts on stream water quality across Taiwan; the results showed that high degrees of landscape fragmentation and interspersion were correlated with the degradation of water quality in terms of high biological and chemical oxygen demands and nitrogen loads. Such significant relationships are found in certain land-use types as pollution sources [18]. Various studies have established a nexus between human activity and increases in edge density (and hence in Shannon's diversity index (SHDI)) with decreases in water quality [19]. Other studies have correlated the patch index and patch density with surface water quality [5,9,20], showing the loss of filtering capacity of fragmented forests relative to non-fragmented forests [21]. Therefore, LULC and landscape pattern analysis can provide important indicators of water quality fluctuations, indirectly reflecting human activity and pollutant transmission [5,22].

Various studies have investigated the impact of land use patterns on water quality. Bu et al. [23] used a stepwise multiple regression to analyze the impact of land use and landscape indicators on surface water quality in the Taizi River Basin in China. Similar studies have also been conducted to assess the impact of different scales of land use. Stone et al. [9] investigated the impact of land use on water quality at three different scales, including reach, bank, and catchment scales. Chiang [18] used landscape metrics to assess the impact of LULC on stream water quality in Taiwan. In addition, land use patterns have a significant impact on groundwater quality [24]. Multivariate analysis was also applied to study the relationship between land use patterns and water quality in different regions. Cluster analysis can classify areas by land use patterns and water quality indicators, allowing researchers to investigate their relationships by group [25]. A redundancy analysis (RDA) was used to investigate the relationship between two sets of variables and has been shown to be effective in assessing the impact of land use on water quality [9].

Although many studies have been conducted to investigate the effect of watershed scales on water quality [26–28], these results have not been consistently recognized due to differences in geographic location and study design. Few studies have compared the spatial variability of the relationship between landscape indicators and water quality in the Dianchi Lake Basin. In addition, most studies used multivariate analysis methods [29], but few studies used modified coupled models and geographic detector methods for quantitative analysis.

The objectives of this study are to: (1) analyze the water-quality changes in Dianchi Lake from 1990 to 2020, using the CWQI (Canadian Water Quality Index) to evaluate the temporal and spatial changes in Dianchi Lake water quality; (2) analyze the relationship between the landscape index and Dianchi Lake water quality through the RDA model and multiple regression analysis, so as to explore the LULC; (3) construct a coupled model of the landscape index and water-quality system and combine it with geographic detectors to analyze the decisive influencing factors. The research results can guide the comprehensive planning, governance, and protection of the watershed, and provide a scientific basis for the sustainable development of the watershed.

## 2. Materials and Methods

Figure 1 shows the conceptual framework of this study, which was divided into five stages: (1) satellite data collection and land-use classification to calculate the landscape pattern metrics; (2) quality analysis of Dianchi Lake water using CWQI; (3) testing the relationship between landscape pattern and water quality based on a redundancy analysis (RDA) and stepwise multiple linear regression (SMLR); (4) constructing a modified coupling model of the landscape pattern and water-quality system; and (5) using geographic detectors to analyze the impact coordination determinant factor.

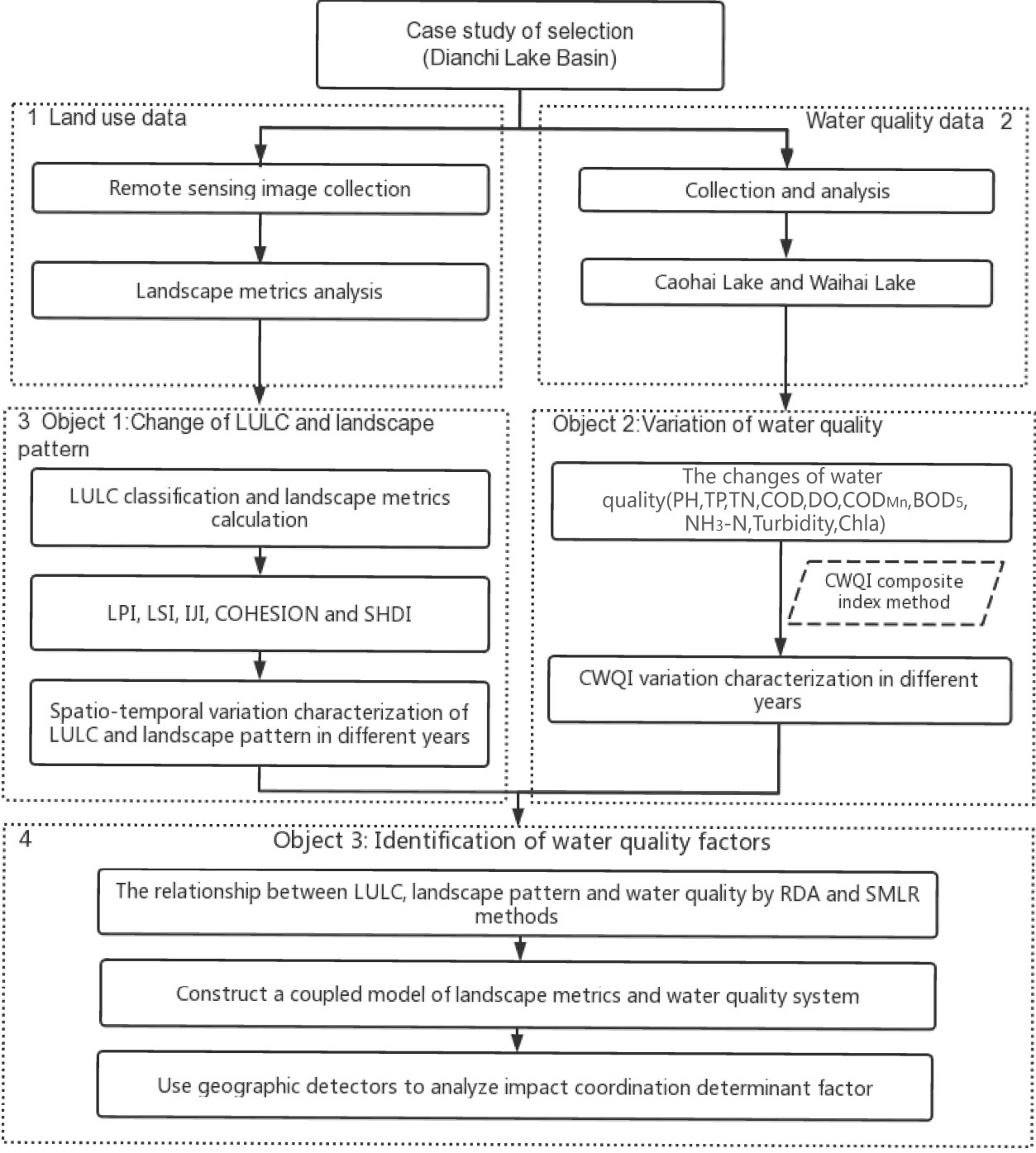

**Figure 1.** A flow chart showing the study's conceptual framework.

### 2.1. Study Area

The Dianchi Lake Basin (24.46° N–25.46° N, 102.5° E–103.0° E) is located in the Yunnan–Guizhou Plateau in Yunnan Province (Figure 2). It is a part of the Jinsha River system in the upper reaches of the Yangtze River Basin, which covers an area of 2920 km$^2$ and is located at an elevation of approximately 1890 m. The main body of water in the Dianchi Basin is an inland plateau lake of the Yangtze River system, which is the largest freshwater lake in the Yunnan–Guizhou Plateau. This basin includes rivers (36 rivers flowing into and out of Yunnan), and its major land-use types can be divided into forest (semi-humid, evergreen, broad-leaved forests), fields (fertile land in the east bank), shrub lands, built-up lands, farmlands, lakes (309 km$^2$ of water surface), and grasslands (the surrounding Yunnan marsh meadow wetland), amongst others.

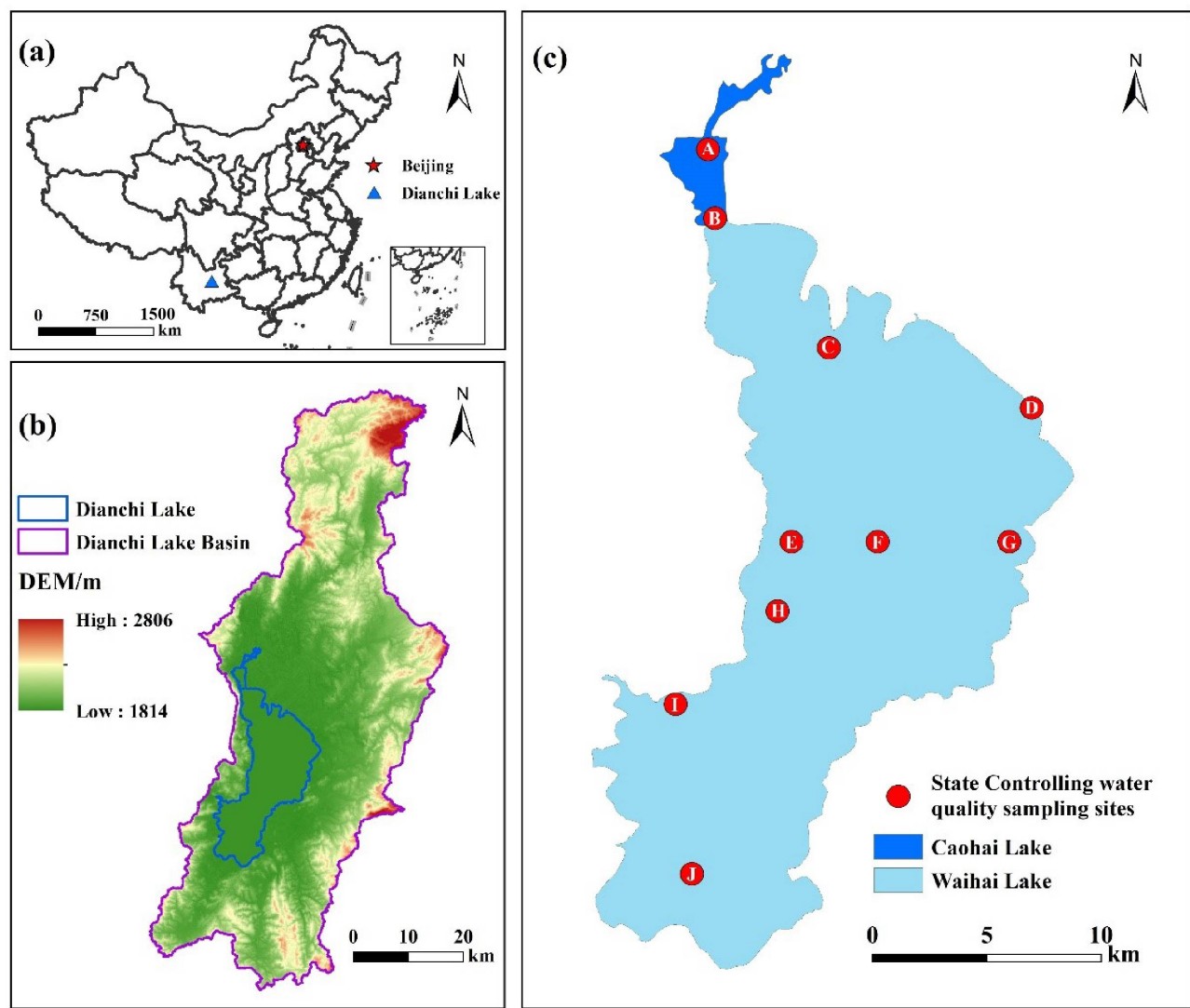

**Figure 2.** Geographical location of the study area. Note: the location of Dianchi Lake in China (**a**); the elevation of Dianchi Lake Basin (**b**); and the location of 10 water quality monitoring sampling sites in Dianchi Lake (**c**). The map images originate from Environmental Sciences, Chinese Academy of Sciences (http://www.resdc.cn/, accessed on 13 September 2022).

Dianchi Lake was once an important source of drinking water for Kunming, the capital of Yunnan Province. Its watershed covers 0.75% of Yunnan Province's land area, supporting 8% of its population and 23% of its gross domestic product (GDP). It has been affected by rapid urbanization, industrial growth, and the intensification of agricultural

activities [30]. The Dianchi Lake Basin has experienced serious water supply and water quality problems [31]. The Dianchi Lake Basin is a resource water shortage area. For many years, the available water resource has been $1.0 \times 109$ m$^3$, which indicates that the water resources per capita are only approximately 300 m$^3$ per year (only 13% of the national average) [32]. The pollution load of nitrogen and phosphorus in farmland runoff has become the main source of pollution in the basin and is an important cause of the eutrophication of Lake Dianchi [33,34]. In 1996, the construction of the Xiyuan Tunnel stopped the flow of the heavily polluted Lake Caohai into Waihai. Lake Dianchi was divided into two sub-lakes: the smaller Caohai Lake in the north and the larger Waihai Lake in the south. The water surface of Caohai Lake only accounts for 3.6% of the area of Lake Dianchi, and its water depth is 1–3 m. Waihai Lake, is the main body of Lake Dianchi, with an area of 289.2 km$^2$ and an average water depth of 5 m [35–37]. Caohai Lake and Waihai Lake each have artificial locks to control their water flow northward to the Tanglangchuan, Pudue, and Jinsha Rivers. In light of the substantial difference in water quality between Caohai and Waihai Lake and the differing impacts of urbanization, Dianchi Lake was divided into Caohai Lake and Waihai Lake in our study.

### 2.2. Data Sources

### 2.2.1. Land-Use Data Acquisition and Processing

The remote sensing image data of the Dianchi Lake Basin for every year from 1990 to 2020 was downloaded from the geospatial data cloud (http://www.gscloud.cn, accessed on 22 July 2021), and the land use information was extracted by using the ENVI5.3 software, based on the sample-oriented, object-oriented, supervised classification method. According to the national classification standard of land use status (GB/T 210-2017), combined with the actual situation of land use in the Dianchi Lake Basin and the operability of interpretation, the land-use types were divided into eight categories: high forest land, shrub land and open forest land, grassland, farmland, water, wasteland, and construction land. With the help of the actual land-use data collected and recorded in the field, accuracy was verified using a confusion matrix. The Kappa coefficient was 0.91, which met the research accuracy requirements. The results of the land-use changes are shown in Figure S1.

### 2.2.2. Water Quality Data

In this study, the monthly water quality data from 1990 to 2020 at ten sampling sites in Caohai and Waihai Lake were used. The average values of the two sampling points in Caohai Lake and the eight sampling points in Waihai Lake were used to characterize the interannual variation in pollutants.

All data were obtained from the Center for Environmental Monitoring in Kunming, Yunnan Province of China (Tables S1 and S2). The data comply with the requirements of the "National Water Quality Sampling Technical Guidelines" (HJ494-2009) and the long-term monitoring specifications for aquatic ecosystems. According to the main pollutants in the Dianchi Lake Basin, we selected ten indicators to characterize the water quality of Dianchi Lake: PH, total phosphorus (TP), total nitrogen (TN), chemical oxygen demand (COD), dissolved oxygen (DO), permanganate index (COD$_{Mn}$), five-day biochemical oxygen demand (BOD$_5$), ammonia nitrogen (NH$_3$-N), turbitidy, and chlorophyll-a (Chla).

### 2.2.3. Landscape Metrics

Changes in landscape patterns result in changes in spatial landscape variables, which affect changes in ecosystem structure and composition [38]. Landscape indexes describe changes in landscape patterns and are useful for characterizing relationships between landscape patterns and landscape change processes [39]. Using actual land-use data collected in the field, a confusion matrix was built to verify the accuracy of the data. The Kappa coefficient was 0.91, meeting the research precision needs. The selected landscape index was calculated using FRAGSTATS 4.2 software (Table S3).

The landscape level can reflect the overall structural characteristics of the landscape [40]. In this study, the largest patch index (LPI), landscape shape index (LSI), interspersion juxtaposition index (IJI, patch cohesion index (COHESION), and Shannon's diversity index (SHDI) were selected (Table 1).

**Table 1.** List of selected landscape metrics at the class and landscape levels in Dianchi Basin.

| Structural Category | Landscape Metrics | Abbreviation | Description |
|---|---|---|---|
| Area | Largest patch index | LPI | Area of the largest patch (unit: %). |
| Shape | Landscape Shape Index | LSI | The complexity of landscape structure (unitless). |
| Isolation and Interspersion | Interspersion Juxtaposition Index | IJI | Proximity of patches in each class. High values correspond to the proportionate distribution of patch type adjacencies (unit: %). |
| Connectivity | Patch Cohesion Index | COHESION | Increases as the patches of the corresponding patch type become less connected (unit: %). |
| Diversity | Shannon's diversity index | SHDI | A measure of diversity in community ecology; indicates the patch diversity in a landscape (unitless). |

### 2.3. Methods

#### 2.3.1. Index System Construction

In this paper, the quantitative indexes of watershed water quality and the landscape pattern system were divided into two categories: landscape metrics and water quality (Table 2).

**Table 2.** Evaluation index system of water quality and landscape pattern in Dianchi Lake Basin.

| Standard Layer | Number | Index | Category | Weight | |
|---|---|---|---|---|---|
| Landscape metrics | L1 | LPI | – | 46.18% | |
| | L2 | LSI | – | 12.51% | |
| | L3 | IJI | + | 5.56% | |
| | L4 | COHESION | – | 20.65% | |
| | L5 | SHDI | – | 15.10% | |

| Standard Layer | Number | Index | Category | Waihai Lake Weight | Caohai Lake Weight |
|---|---|---|---|---|---|
| Water quality index | W1 | PH | neutral | 9.63% | 17.78% |
| | W2 | TP | – | 7.27% | 15.53% |
| | W3 | TN | – | 5.59% | 9.70% |
| | W4 | COD | – | 14.37% | 3.65% |
| | W5 | DO | + | 16.75% | 16.89% |
| | W6 | $COD_{Mn}$ | – | 14.71% | 7.14% |
| | W7 | $BOD_5$ | – | 8.15% | 6.43% |
| | W8 | $NH_3$-N | – | 5.33% | 8.80% |
| | W9 | Turbidity | + | 11.24% | 8.20% |
| | W10 | Chla | – | 6.93% | 5.89% |

#### 2.3.2. Redundancy Analysis

The relevant indicators with a significant impact of landscape structure on water quality and their corresponding optimal scales were screened through a redundancy analysis (RDA). An RDA is a constrained ranking of a principal component analysis (PCA) and can be used for the multivariate statistical analysis of multiple explanatory variables and response variables [12]. Taking the LULC and landscape indexes as explanatory variables and the ten water quality indicators as response variables, an RDA was run to analyze the correlation between the water quality and landscape indexes. A total of 999 Monte Carlo permutation tests were used to analyze the significance of explanatory variables. The explanation rate of each scale's landscape index on water quality were then screened. The above statistical analysis was completed using Canoco 5.0 software.

### 2.3.3. Canadian Water Quality Index (CWQI)

The Canadian Water Quality Index (CWQI) method calculates whether the water-quality monitoring data exceed the water quality standard limit in three aspects: scope, frequency, and amplitude [5]. The index provides a mathematical framework for assessing environmental water quality conditions associated with water quality objectives [41,42] (Table S4).

### 2.3.4. Statistical Analysis

We used the Pearson correlation coefficient to test the correlation between each landscape index and each water quality index. We then applied a redundancy analysis (RDA) and stepwise multiple linear regression (SMLR) to analyze the correlation between the land cover type change and water quality [5,40].

### 2.3.5. Entropy–TOPSIS Model

The comprehensive situation of the landscape pattern and water quality of Dianchi Lake basin from 1990 to 2020 was evaluated year by year in this paper using the entropy—TOPSIS method. The entropy method can be used to determine the relative importance of different factors in affecting a focal variable that avoids the subjectivity of an artificial weight assignment [39]. The weights of the water quality index and landscape pattern index are shown in Table 2. The TOPSIS method is an evaluation method based on multi-objective decision making, wherein positive ideal solutions and negative ideal solutions of each index in the target are identified and the distance between the index and the positive and negative ideal solutions is measured (i.e., the closeness degree). The entropy–TOPSIS model is a combination of the entropy weight method and the TOPSIS method, and the details of the calculation method are provided in a previous study [43].

### 2.3.6. Modified Coupling Coordination Degree Model

With a deepening understanding of the scientific outlook on development, the coupling coordination degree model has become an effective evaluation and research tool for the regional overall balanced development. Coupling refers to the phenomenon in which two or more systems or motion modes affect each other or even unite through various interactions. However, for this model, there are four types of misuses, including writing errors, coefficient loss, weight misuse, and model failures, which have significantly affected the scientific nature of academic research. We used the modified coupling coordination degree model, which has better validity [44].

To characterize the development of the water quality and landscape pattern in the Dianchi Basin, a modified coupling degree model of water quality and landscape patterns was built using the coupling principle of physics, which provides a quantitative measure of the coupling degree of the water quality and landscape pattern systems. The calculation formula is as follows [45]:

$$f(x) > f(yn), C = \sqrt{\{1 - [f(x) - f(yn)]\} \times f(yn)/f(x)}$$
$$f(x) < f(yn), C = \sqrt{\{1 - [f(yn) - f(x)]\} \times f(x)/f(yn)}$$

where $f(x)$ represents the comprehensive evaluation value of the Dianchi Lake Basin landscape metrics based on the entropy–TOPSIS model, $f(yn)$ represents the comprehensive evaluation value of Caohai Lake or Waihai Lake water quality, based on the entropy–TOPSIS model, and $C$ represents the coupling degree. The interval for $f(x)$, $f(yn)$, and $C$ is [0, 1]. Based on this modified model, the coupling degree can more reasonably represent the coupling coordination relationship and development level between the landscape pattern and the water quality of Dianchi Lake.

As the coupling degree places more emphasis on the interaction degree between subsystems, it does not effectively prove whether each coupling state exhibits benign coordination. Therefore, the coupling coordination degree model was introduced to analyze

the coordination status of the interaction coupling between the water quality and the landscape pattern systems of the Dianchi Lake Basin. The calculation formula is [45]:

$$D = \sqrt{C \times T}$$
$$T = af(x) + bf(yn)$$

where $D$ is the coupling coordination degree and $T$ is the comprehensive development level. In this paper, the water quality and the land-use system were considered equally important in the urbanization of Dianchi Lake Basin. So, both $a$ and $b$ take 1/2 [46]. The criteria for evaluating the coupling degree and coordination degree were determined based on previous studies and current conditions [46–48] (Table S5).

In this research, C1 is the coupling degree of Waihai Lake and C2 is the coupling degree of Caohai Lake. The coordinated development of the landscape pattern and water-quality system in Dianchi Basin was determined via calculations. D1 is the coupling coordination degree of Waihai Lake and D2 is the coupling coordination degree of Caohai Lake.

2.3.7. Geographical Detector Model

Geographic detector models are used to reveal the differences in the spatial distribution of geographical objects and the forces driving them. The factor detector model can identify the independent variables that affect water quality and landscape patterns. The formula is as follows [49]:

$$q = 1 - \frac{\sum_{h=1}^{L} N_h \sigma_h^2}{N \sigma^2}$$

where $q$ is the measure of the explanatory power of the independent variable, $L$ is the stratification of independent variables, $N_h$ and $\sigma_h^2$ are the number of cells and the variance of layer H, respectively, and $N$ and $\sigma^2$ are the unit number and the variance of the study area overall, respectively.

Since the application of the geographic detector requires the argument to be a typed quantity, it is necessary to convert the numeric quantity into a typed quantity. In this study, the optimal parameter discretization and the realization of factor detector results were carried out according to the q value using the software RStudio GD package [50].

A geographical detector model with the coupling coordination degree as the dependent variable and the water quality and the landscape pattern system as the independent variable was built to determine the main indicators affecting the variations in coupling coordination degree in Caohai Lake and Waihai Lake.

## 3. Results

### 3.1. Dynamic Changes in Land Use in Dianchi Lake Basin

The areas of farmland, high forest land, shrub land, and open forest land decreased by 332.30 km$^2$, 70.85 km$^2$, 46.71 km$^2$, and 15.24 km$^2$, respectively. The areas of construction land, grassland, wasteland, and water body increased by 392.13 km$^2$, 34.59 km$^2$, 30.43 km$^2$, and 9.74 km$^2$, respectively. In particular, farmland area was decreased from 26.90% in 1990 to 14.68% in 2020, and the built land was increased from 8.05% in 1990 to 21.98% in 2020. The major land-use types in the Dianchi Lake Basin in 1990 were forest land, water, and farmland; in 2020, the major land-use type became built land.

The spatial distribution of land-use types (except Dianchi water) in the Dianchi Basin showed that it became built land from farmland in the central Dianchi Lake Basin, and the forest land is mainly distributed in the northern and southern parts of the basin.

The change in the landscape pattern indexes from 1990 to 2020 is dramatic (Figure 3). The LPI first decreased and then increased, showing a downward trend from 1990 to 2009, and began to increase rapidly in 2010. The LSI continued to increase, indicating a more complex plaque shape. LJI has shown a trend of continuous increase, from 73.95 in 1990

to 80.46 in 2020, with a smaller fluctuation range. COHESION measured the aggregation of different patch types in the landscape, showing a continuous decreasing trend, with smaller values reflecting less aggregation. SHDI showed a slow increasing trend. These changes substantially altered the spatial distribution characteristics of the original natural landscape and resulted in a decrease in landscape dominance.

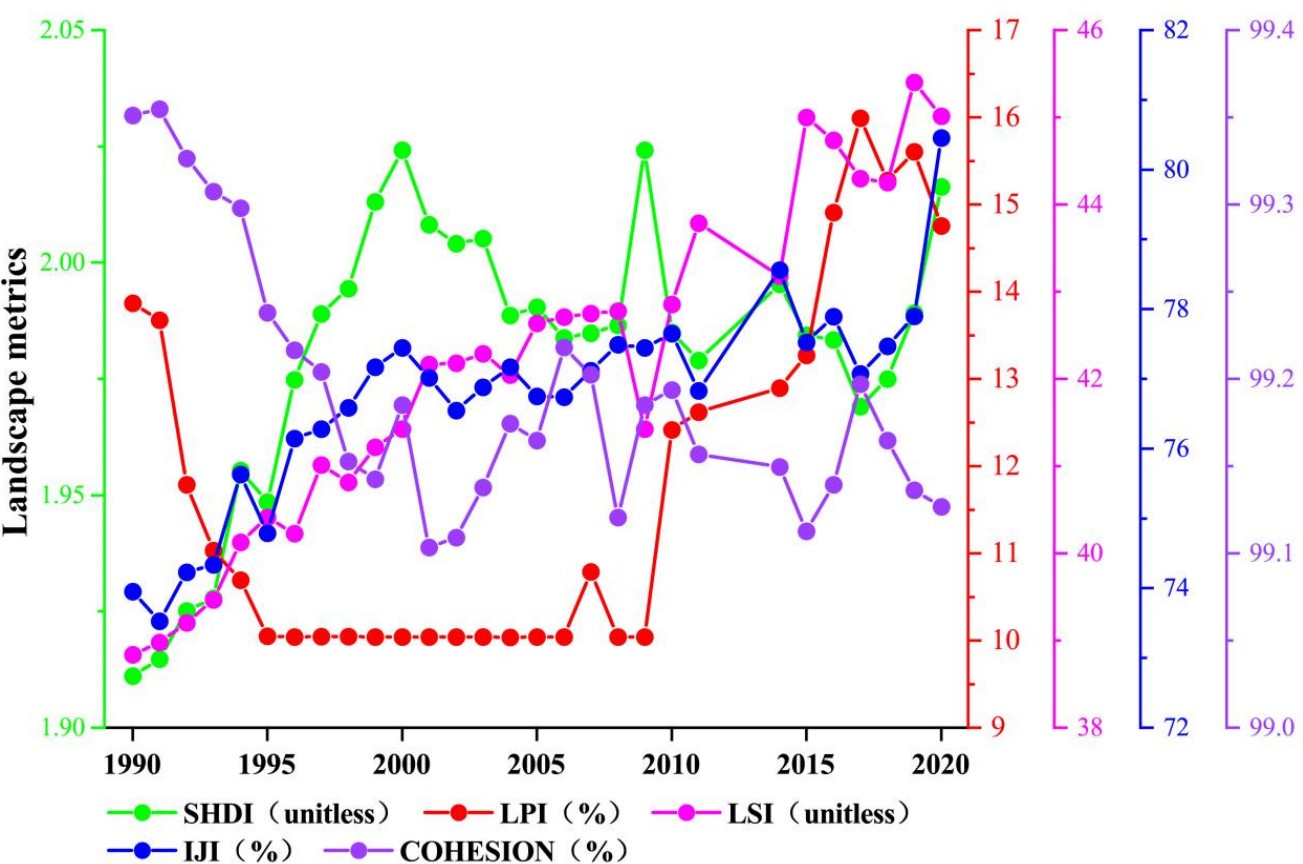

**Figure 3.** Changes in landscape pattern indexes in Dianchi Lake Basin.

*3.2. Temporal and Spatial Variation of Water Quality in Dianchi Lake*

Spatial differences in the water quality of Dianchi Lake were observed in the concentrations of TN, TP, $COD_{Mn}$, DO, and $BOD_5$ between Caohai Lake and Waihai Lake (Figure 4). The average total nitrogen in Waihai Lake was 1.92 mg/L during the 30 year period, and in Caohai Lake it was 8.67 mg/L. The average total phosphorus during the 30year period was 0.16 mg/L in Waihai Lake and 0.76 mg/L in Caohai Lake, 4.75 times higher than the average total phosphorus in Waihai Lake. The average biochemical oxygen demand was 3.91 mg/L and 10.98 mg/L in Waihai and Caohai Lake, respectively, and the oxygen demand was thus 2.8 times higher in the latter than in the former. The water velocity of Caohai Lake is low, and the diffusion of pollutants is slow. The velocity of water in Waihai Lake is high, and the diffusion of pollutants is rapid. Regarding Caohai Lake, the water quality first decreased and then increased. From 1990 to 2000, TN, TP, and DO concentrations increased. From 2000 to 2014, the water quality increased slowly, and the values of each index decreased slowly. The water quality has increased significantly since 2014. Most of the pollutants produced in the main city of Kunming flow into Caohai Lake, which has a smaller water area and is more seriously polluted than Waihai Lake.

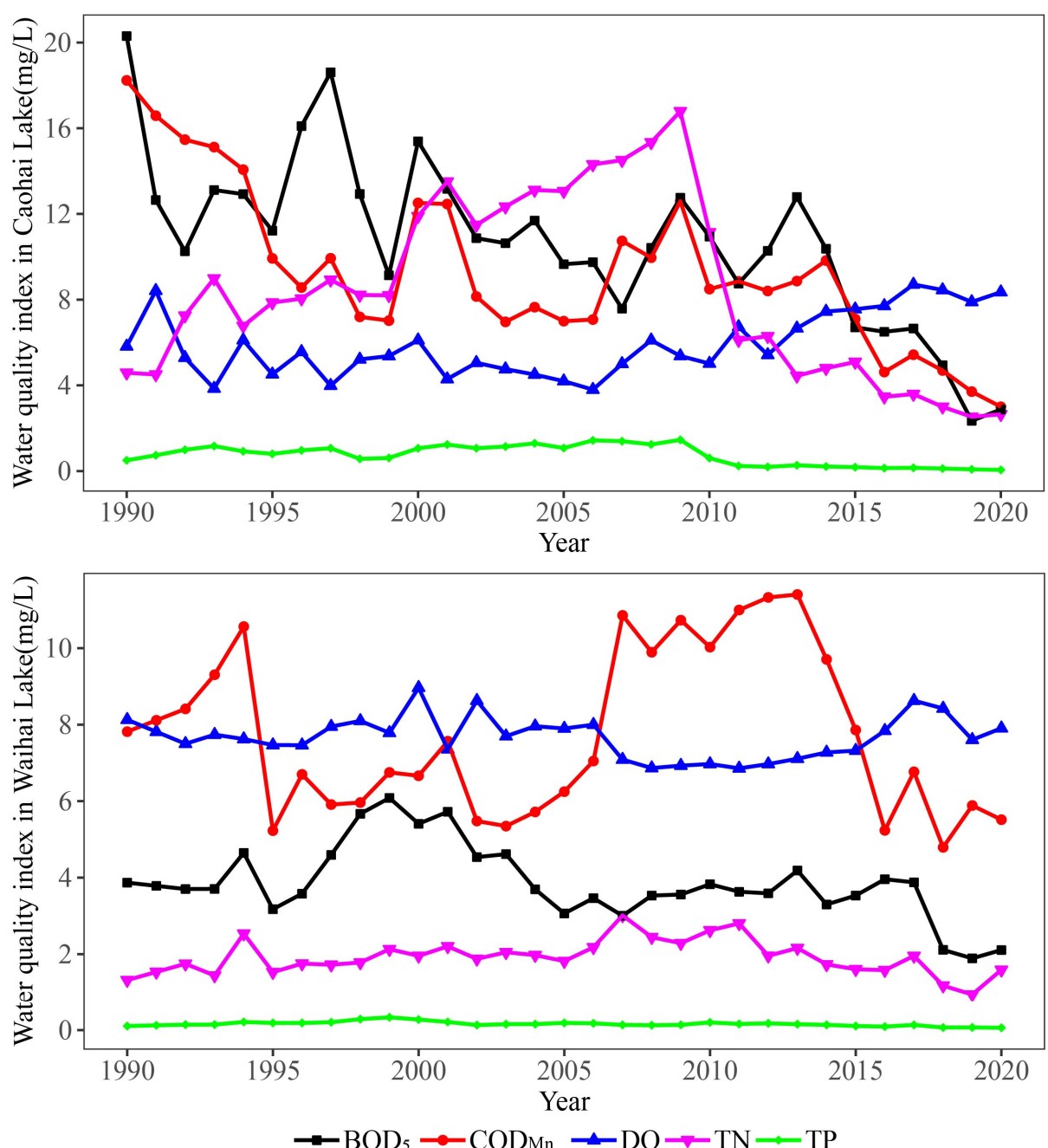

**Figure 4.** Changes in water quality indexes of Dianchi Lake.

The water quality in Waihai Lake first decreased and then increased. Water quality has improved significantly since 2013. In 1999, Kunming carried out the "zero-point action" campaign to meet the standards set for the discharge of Dianchi Lake. A sewage treatment plant and sewage pump station were established thereafter. Various measures have been taken to enhance the treatment of Dianchi Lake, and the results of these measures have been positive.

### 3.3. CWQI Temporal Variations of Dianchi Water

The water quality of Caohai Lake and Waihai Lake jointly determines the water quality of the entire Dianchi Lake Basin. From 1990 to 2020, the CWQI values for the entire Dianchi Lake waters ranged from 64.72 to 92.38 (73.38 + 7.28). The Caohai Lake CWQI values ranged from 49.10 to 88.23 (59.93 + 11.68), and the Waihai Lake values ranged from 49.10 to

88.23 (59.93 + 11.68). In the same year, the CWQI values of Caohai Lake were significantly lower than those of Waihai Lake (Figure 5).

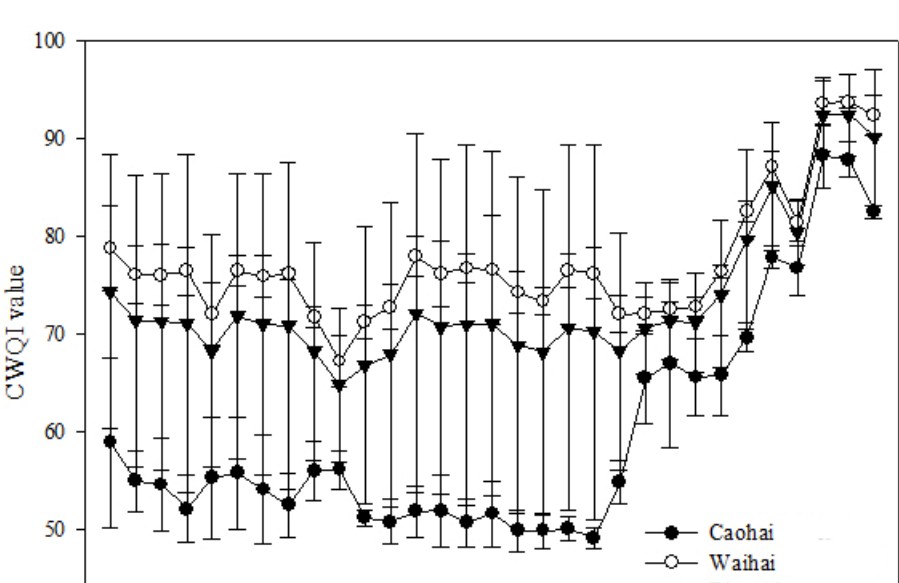

**Figure 5.** Changes in CWQI in Waihai Lake, Caohai Lake, and Dianchi Lake from 1990 to 2020.

The CWQI value of the Dianchi Lake waters demonstrates obvious time points. From 1990 to 2020, the change of CWQI value in Caohai Lake can be divided into two time periods: 1990–2009 and 2010–2020. From 1990 to 2009, the CWQI value of Caohai Lake fluctuated and decreased. The maximum CWQI value in 1990 was 58.90. The minimum value was 49.10, of which the fluctuations were large between 1990 and 2000 (between 52.06 and 58.90), and the fluctuations between 2000 and 2009 were small (between 49.10 and 51.92). The CWQI value of Caohai Lake increased significantly from 2010 to 2020. It increased to 82.47 in 2020, reaching its largest value (88.23) in 2018, and exceeded 80.00 in the period of 2018–2020.

The change in CWQI value in Waihai Lake was different from the change in Caohai Lake. From 1990 to 1999, the CWQI value in Waihai Lake continued to decrease, from 78.73 in 1990 to 67.18 in 1999, and the water quality in the outer sea fell to its worst level. With fluctuating changes (71.23–77.88), the water quality of Waihai Lake improved significantly from 2011 to 2020; the CWQI value increased almost linearly, only decreasing in 2017 (81.35). Since 2015, the CWQI value of Wahai Lake has exceeded 80.00, and it has been high since its 2018 value of 90.00. The change in trend of the CWQI value for the entire Dianchi Lake water content is consistent with Waihai Lake. The CWQI value has exceeded 80.00 since 2016, and it exceeded 90.00 in the 2018–2020 period.

*3.4. Correlation Analysis amongst LULC, Landscape Patterns, and Water Quality in Dianchi Lake Basin*

The relationships among the LULC, landscape pattern, and water quality in Waihai Lake and Caohai Lake were different (Figure 6). Assessed by sampling water plots, the first two axes accounted for approximately 50.20–72.70% of water-quality changes, with a minimum of approximately 14.30% in Waihai Lake and a maximum of 47.50% in Caohai Lake (Table 3). The pollution gradient (a positive correlation with COD and a negative correlation with DO, PH, and turbidity) in the first axis correlated negatively with Others.

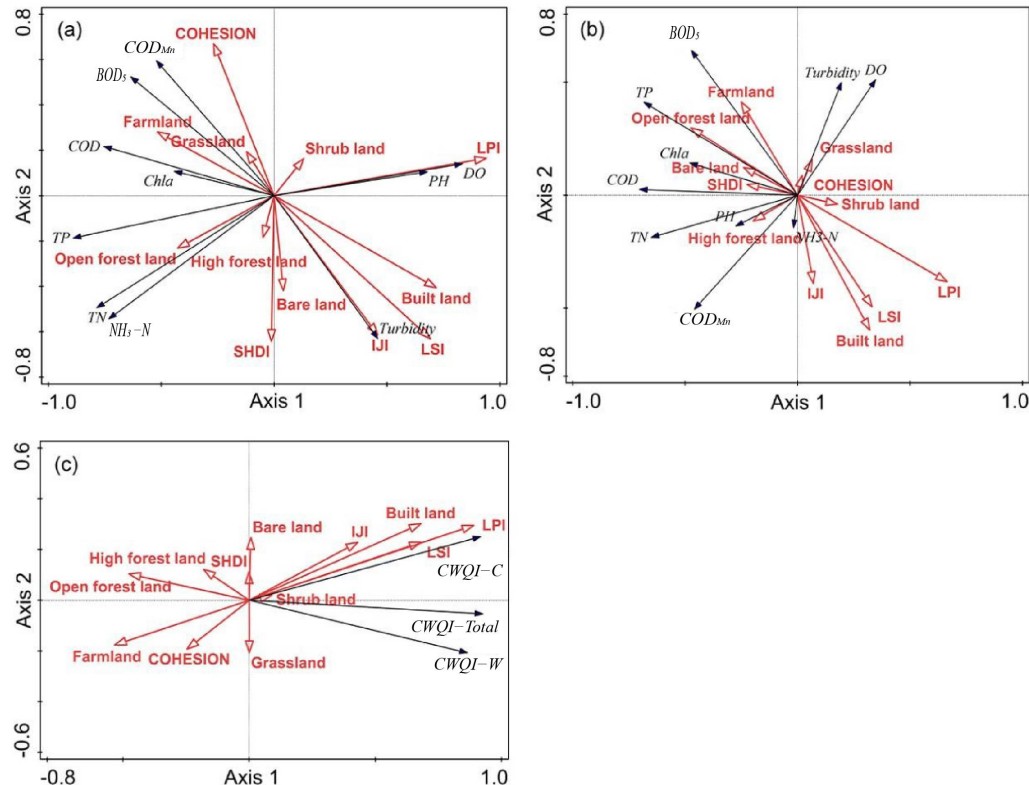

**Figure 6.** The correlation between land use variables, landscape metrics (represented by red arrows) and water-quality parameters (represented by black arrows) for Caohai Lake (**a**), Waihai Lake (**b**), and CWQI (**c**) using the RDA. Red arrows represent land use and landscape pattern indices; black arrows represent water quality. The length of the arrow indicates the contribution of the metric to the ranking model. The shorter the arrow, the smaller the contribution, and vice versa. The cosine of the angle between the arrows represents the correlation between the indicators.

**Table 3.** The RDA results for the percentage of overall water-quality parameters are explained by the explanatory variables.

| Water Type | Explained Variation (%) | | | Pseudo-F | *p* Value | Explanatory Variables (Contribution %) |
|---|---|---|---|---|---|---|
| | Axis 1 | Axis 2 | All Axes | | | |
| Caohai | 47.55 | 16.9 | 72.7 | 4.9 | 0.002 ** | LPI (55.7) LSI (19.7) Bare land (4.4) |
| Waihai | 22.81 | 14.37 | 50.2 | 2.0 | 0.004 ** | LPI (23.5) Built land (11.1) LSI (10.2) Bare land (9.0) |
| CWQI | 80.13 | 2.72 | 82.84 | 8.6 | 0.002 ** | LPI (76.0) COHESION (7.0) High forest land (4.1) |

Note: ** Extremely significant correlation ($p < 0.01$).

In Caohai Lake, built land and bare land correlated positively with turbidity; open forest land correlated positively with TP, TN, and $NH_3$-N; farmland, grassland, and COHESION correlated positively with the $BOD_5$, COD, $COD_{Mn}$ and Chla; and shrub land and LPI were positively related with PH and DO. The relation between water quality and landscape metrics in Waihai Lake was different compared to Caohai Lake. Grassland and COHESION correlated positively with turbidity and DO; farmland, open forest land, bare land, and SHDI correlated positively with the BOD, TP, COD, and Chla; and high forest land correlated negatively with the TN, $COD_{Mn}$, PH, and $NH_3$-N.

In Caohai Lake (Figure 6a), the LPI, IJI, LSI, and built land COHESION arrows are longer, which indicates that they have a great influence on water quality. In Waihai Lake, (Figure 6b), the indicators that have a greater impact on water quality are the LPI, LSI, and built land.

The most important explanatory variables in Caohai Lake were LPI (55.7%), LSI (19.7%), and bare land (4.4%). The most important explanatory variables in Waihai Lake were the LPI (23.5%), built land (11.1%), LSI (10.2%), and bare land (9.0%). The largest explanatory variable was LPI for Caohai Lake and Waihai Lake (Table 3). In Caohai Lake (Figure 5a), the LPI, ILI, LSI, and built-land were positively correlated with the PH, DO, and Turbidity, and were negatively correlated with the $COD_{Mn}$, $BOD_5$, COD, Chla, TP, TN, and $NH_3$-N.

The RDA axis explained less variation in Waihai Lake than Caohai Lake (Table 2; Figure 6). In Waihai Lake, the forest land, farmland, bare land, high forest land, and SHDI showed a positive correlation with the $BOD_5$, TP, Chla, COD, TN, PH, and $COD_{Mn}$; the $BOD_5$, TP, Chla, and COD showed a negative correlation with the LPI, ILI, LSI, and built-land.

The correlation of the CWQI with the LULC and landscape metrics was varied between Caihai Lake, Waihai Lake, and the total water of Dianchi Lake (Figure 6c). The built land, LPI, LSI, IJI, shrub land, and CWQI of Caohai Lake correlated well, though the arrow for the shrub land was shorter. The CWQI relations of Waihai and the total water were less than for Caohai Lake: only grassland was positively correlated with the CWQI of Waihai Lake (Table 4). Regarding the CWQI, the most important explanatory variables were the LPI (76.0%), COHESION (7.0%), and high forest land (4.1%). This observation was supported by the results of SMLR (Table 4), suggesting that the LULC and landscape pattern have a greater impact on Caohai Lake's water quality.

**Table 4.** Stepwise multiple linear regression (SMLR) of LULC, landscape metrics, and water-quality parameters.

| Area | Water-Quality Parameters | $R^2$ | Adjusted $R^2$ | F | Significance | Related to LULC | Related to Landscape Metrics |
|---|---|---|---|---|---|---|---|
| Caohai Lake | PH | 0.406 | 0.385 | 19.800 | 0.000 ** | / | LPI (+) |
| | TP | 0.756 | 0.738 | 43.292 | 0.000 ** | Bare land (−) | LPI (−) |
| | TN | 0.663 | 0.651 | 57.061 | 0.000 ** | / | LPI (−) |
| | COD | 0.602 | 0.573 | 21.144 | 0.000 ** | / | LPI (−), LSI (−) |
| | DO | 0.712 | 0.702 | 71.554 | 0.000 ** | / | LPI (+) |
| | $COD_{Mn}$ | 0.673 | 0.662 | 59.677 | 0.000 ** | / | LSI (−) |
| | $BOD_5$ | 0.700 | 0.679 | 32.723 | 0.000 ** | High forest land (−) | LSI (−) |
| | $NH_3$-N | 0.744 | 0.705 | 18.919 | 0.000 ** | Open forest land (+) | LPI (−), LSI (+), COH-ESION (+) |
| | Turbidity | 0.435 | 0.416 | 22.363 | 0.000 ** | / | LSI (+) |
| | Chla | 0.224 | 0.198 | 8.388 | 0.007 ** | / | LPI (−) |
| | CWQI–C | 0.877 | 0.868 | 100.025 | 0.000 ** | / | LPI (+), COHESION (−) |
| Waihai Lake | PH | 0.135 | 0.105 | 4.532 | 0.042 * | Bare land (+) | / |
| | TP | 0.432 | 0.412 | 22.035 | 0.000 ** | / | LPI (−) |
| | TN | 0.192 | 0.164 | 6.887 | 0.014 * | / | LPI (−) |
| | COD | 0.347 | 0.324 | 15.405 | 0.000 ** | / | LPI (−) |
| | $BOD_5$ | 0.273 | 0.248 | 10.898 | 0.003 * | / | LPI (−) |
| | $NH_3$-N | 0.199 | 0.171 | 7.192 | 0.012 * | / | IJI (+) |
| | Turbidity | 0.159 | 0.13 | 5.479 | 0.026 * | Built land (−) | / |
| | CWQI–W | 0.500 | 0.483 | 29.002 | 0.000 ** | / | LPI (+) |
| | CWQI–Total | 0.709 | 0.688 | 34.051 | 0.000 ** | / | LPI (+) LSI (+) |

Note: * Significant correlation ($p < 0.05$); ** Extremely significant correlation ($p < 0.01$); "+" represents a positive correlation; "−" represents a negative correlation; "/" represents no significance value.

### 3.5. The Temporal Change in the Coordinated Development of the Landscape Pattern and Water Quality in Dianchi Lake Basin

The coupling degrees between Waihai Lake (C1) and Caohai Lake (C2) have been in the running-in stage for a long time. C1 and C2 both showed a trend of increasing from high to low and then fluctuating (Figure 7). Overall, C1 was better than C2 during the same

period, indicating a higher degree of coupling in Waihai Lake. From 1990 to 2004, both C1 and C2 experienced a rapid decline, from high-level coupling to an antagonistic stage, and C1 also experienced low-level coupling between 2002 and 2005. Beginning in 2005, C1 and C2 started to fluctuate and rise; most of the time, they were in the running-in stage. In 2014, C1 entered high-level coupling. In 2017, C2 also entered high-level coupling. However, in 2020 C1 and C2 both returned to pre-1990 levels.

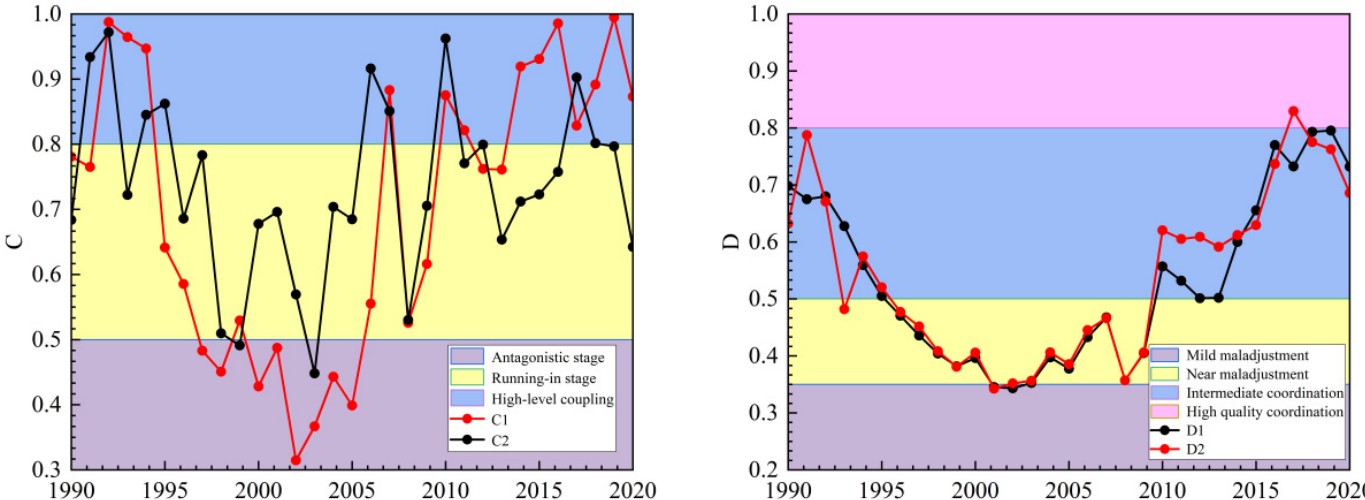

**Figure 7.** Coupling degree variation and coupling coordination degree variation in Caohai Lake and Waihai Lake.

The coupling coordination degrees, D1 and D2, demonstrated a trend of increasing slowly after decreasing continuously and exhibited a U-shaped pattern. From 1990 to 1995, D1 and D2 decreased from intermediate coordination to near maladjustment. From 1996 to 2009, they decreased to mild maladjustment. During this period, the values of D1 and D2 were at their lowest stage, and the lowest value was only 0.34. Beginning in 2010, the coupling coordination degrees (D1 and D2) began to show a fluctuating upward trend, reached a maximum value in 2018 and then decreasing slightly. The final coupling coordination degrees, D1 and D2, are only better than the 1990 level. This shows that the coupling coordination between the landscape patterns and water quality levels of Waihai Lake and Caohai Lake has been in a state of imbalance for a long time. After many years of water-quality control in Dianchi Lake, the interaction between the landscape pattern and the water-quality system in the Dianchi Lake Basin tends to be stronger.

The coupling coordination types of Caohai Lake and Waihai Lake from 1990 to 2020 were divided into basic coordination and moderate quality coordination (Figure 8). Caohai Lake experienced a moderate coordination period from 1990 to 1995, during which the water quality impact lagged behind; between 1996 and 2009, it entered a basic co-ordination period and the landscape pattern impact lagged behind. Between 2010 and 2020, Caohai Lake entered a moderate coordination period, mainly due to the lag of landscape pattern impact. Waihai experienced a moderate coordination period from 1990 to 1995; however, the water quality lagged behind and the landscape just lagged for two years. The basic coordination period was from 1996 to 2000, with the landscape pattern lagging behind. From 2001 to 2002, it was mild maladjustment, and the impact of the landscape pattern was lagging behind; from 2003 to 2009, it returned to the basic coordination period, and the impact of the landscape pattern lagged behind; from 2012 to 2020, it returned to the moderate coordination period, with the main impact on water quality lagging behind.

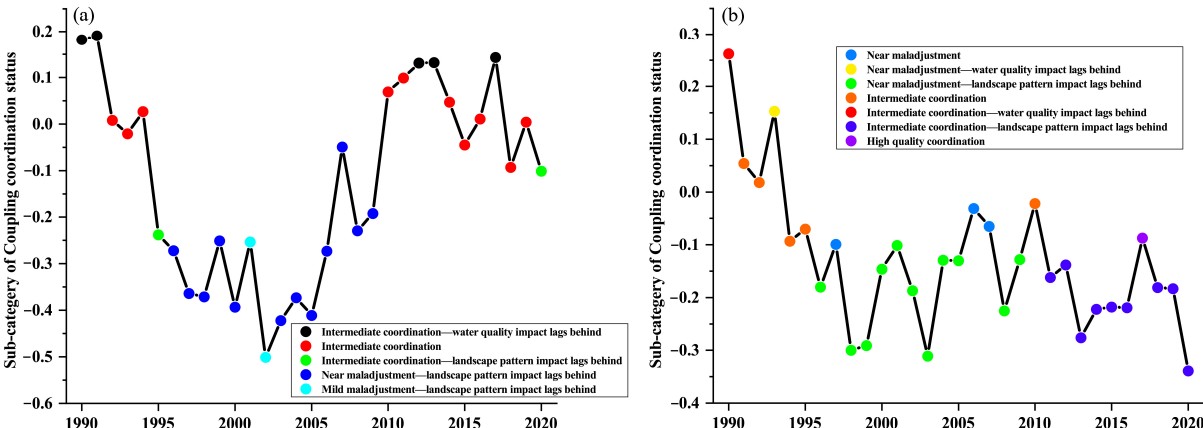

**Figure 8.** Sub-category of coupling coordination status of water quality and landscape metrics in Waihai Lake (**a**) and Caohai Lake (**b**) from 1990–2020.

Overall, Dianchi Lake returned to a moderate-quality coordination in 2020, and both Caohai and Waihai Lake are dominated by lagging landscape pattern impact.

### 3.6. Identification of Indicators Affecting the Coupling Coordination between Landscape Patterns and Water Quality

The detection results of the geographical detector model showed that there are differences in the indicators affecting the coupling coordination between the water quality and the landscape in Caohai Lake and Waihai Lake (Table 5). Among the q values of Caohai Lake, LPI (0.890), TN (0.849), NH$_3$-N (0.831), LSI (0.769), and COD$_{Mn}$ (0.744) demonstrated a strong determining force on the coupling coordination degree. COHESION, SHDI, Chla, IJI, and Turbidity were the five indicators with the lowest q value, ranging from 0.230 to 0.409. The *p* values of the LPI, LSI, TP, TN, COD, DO, COD$_{Mn}$, and NH$_3$-N all reached a significant level below 0.01, indicating that they have a strong significance for the coupling coordination degree of Caohai Lake. Among the q values for Waihai Lake, LPI (0.933), LSI (0.880), COD (0.831), LSI (0.769), BOD$_5$ (0.664), and TP (0.605) demonstrated stronger effects on the coupling coordination degree. The Chla, PH, NH$_3$-N, DO, and COHESION were the five indicators with the lowest q value, ranging from 0.105 to 0.239. The *p*-values of the LPI and LSI reached a significant level below 0.01, and the *p*-values of COD and BOD$_5$ reached a significant level below 0.05, indicating that they have a strong significance for the Waihai Lake coupling coordination. Overall, the coupling coordination degree of water-quality indicators to Waihai Lake is not significant.

**Table 5.** The detection results of the main indicators affecting the variation of coupling coordination degree.

| System | Index | Q of Caohai Lake | Explanatory Ranking | Q of Waihai Lake | Explanatory Ranking |
|---|---|---|---|---|---|
| Landscape metrics | LPI | 0.890 ** | 1 | 0.933 ** | 1 |
| | LSI | 0.767 ** | 4 | 0.880 ** | 2 |
| | IJI | 0.396 | | 0.504 * | |
| | COHESION | 0.230 | | 0.239 | |
| | SHDI | 0.364 | | 0.315 | |
| Water quality index | PH | 0.498 | | 0.181 | |
| | TP | 0.707 ** | | 0.605 | 5 |
| | TN | 0.849 ** | 2 | 0.604 | |
| | COD | 0.669 ** | | 0.703 * | 3 |
| | DO | 0.694 ** | | 0.218 | |
| | COD$_{Mn}$ | 0.744 ** | 5 | 0.257 | |
| | BOD$_5$ | 0.470 | | 0.664 * | 4 |
| | NH$_3$-N | 0.831 ** | 3 | 0.213 | |
| | Turbidity | 0.409 | | 0.336 | |
| | Chla | 0.387 | | 0.105 | |

Note: * Significant correlation (*p* < 0.05); ** Extremely significant correlation (*p* < 0.01).

In summary, the landscape metrics are more decisive for the coupling coordination degree. The landscape pattern index of Waihai Lake is more decisive for the coupling coordination degree than that of the Caohai Lake.

## 4. Discussion

The water quality of Dianchi was significantly impacted by the development and construction of Kunming, as most of the pollutants generated in the main urban area of Kunming flow into Caohai Lake. The built land area was dramatically increased, and the farmland area largely decreased due to the acceleration of urbanization from 1990 to 2020. Nearly all of the land-use types tended to converge from 2010, and the correlation of landscape pattern with the CWQI was tight.

The water quality in Waihai Lake first decreased and then increased. From 1990 to 2012, the value of each index fluctuated, peaking in 2012. Water quality has improved significantly since 2013. In 1999, Kunming carried out the "zero-point action" campaign to meet the standards set for the discharge of Dianchi Lake. A sewage treatment plant and sewage pump station were established thereafter. Various measures were taken to enhance the treatment of Dianchi Lake, and the results of these measures have been positive.

The extent of water-quality deterioration in a watershed depends on the type and abundance of LULCs within the region and the spatial distribution and configuration of landscape patterns [13]. RDA and SMLR results validate the effects of LULC and landscape patterns on water quality. This effect varies widely across years and seasons. In addition, there are significant differences between dry and rainy seasons, resulting in significant changes in the relationship between land use, landscape indices, and water quality between seasons and years [51]. Buildings and farmland are the main, non-point source pollution generators that reduce water quality (Figure 6, Table 3), which is consistent with previous findings [9].

Many studies have shown that the water quality within a watershed is closely related to LULC patterns, as natural vegetation has a positive impact on water quality and farmland and built-up areas have a negative impact [23]. Compared with other land use methods, farmland and construction land are more prone to non-point source pollution [5,9]. Although water-quality degradation is closely related to anthropogenic land use, natural vegetation cover can produce good water quality [52]. Forests and grass are considered key factors in reducing and controlling pollutants, inhibiting soil erosion, and transferring sediments to river surface runoff [53].

A number of landscape metrics have been developed to describe landscape patterns and understand spatial heterogeneity [54]. Pratt and Chang found that the impact of landscape features on water quality varies with seasons [8]. Our study shows that the most important explanatory variables for Caohai Lake are the LPI, LSI, and bare land. The most important explanatory variables for Waihai Lake are the LPI, building land, LSI, and bare land. The largest explanatory variable is the LPI for both Caohai Lake and Waihai Lake. Based on these results, the proportion of built-up land and bare land should be controlled, and the proportion and spatial location of woodland and grassland should be optimized to make the watershed sustainable [55]. The LPI and LSI landscape indicators describe the expansion trend or degree of agglomeration of different patch types in the landscape [55]. Studies have shown that landscape indicators (i.e., PD, LPI, and LSI) are closely related to river water quality [9]. This association is consistent with our results.

Although this paper explores the impact of land use and landscape pattern changes in the study area on the water quality of Dianchi Lake over the past 30 years, it can provide some references for the ecological protection and governance of the basin. However, there are still limitations: (1) in different years and seasons, the complex relationship between landscape patterns and water-quality parameters involves complex mechanisms and processes, which require further research; (2) this paper explores the process and relationships of historical change. The use of better models and methods as well as the

simulation and prediction of future changes in the relationship between water quality and landscape under different development scenarios are worthy of further exploration.

## 5. Conclusions

Based on the LULC and water quality in the Dianchi Basin from 1990 to 2020, we analyzed the relationship between landscape index and the water quality of Dianchi Lake, constructed a coupled model of the landscape index and water-quality system, analyzed the decisive influencing factors using geographic detectors, and conclude that:

(1) Changes in the land-use types were obvious and nonlinear. The majority of land became built land in 2020, changing from forest land and farmland in 1990 (except for the Dianchi water). Landscape pattern indexes also indicated that almost all land-use types were scattered from 1990 to 2000, demonstrated a minor change in 2000–2010, and then were gathered from 2010 to 2020;

(2) Changes in the water quality of Dianchi Lake were similar to changes in the landscape pattern. Meanwhile, it lagged behind changes in land use types. The CWQI value was nearly linearly decreased from 1990 to 1998; it had a slight change in 1999–2013, and then quickly increased from 2013 onward;

(3) Land-use types had a significant impact on the water quality in Dianchi, and landscape indicators had different impacts on the water quality of Caohai Lake and Waihai Lake. The LPI, LSI, and bare land were major factors affecting the water quality of Caohai Lake, whereas the LPI, building land, LSI, and bare land were the dominant factors influencing Waihai lake. The LPI was the largest influence factor in both Caohai Lake and Waihai Lake;

(4) There are different indexes affecting the coupling coordination degree of Caohai and Waihai Lake. The landscape pattern indexes of Waihai Lake (mainly LPI and LSI) are more decisive to the coupling coordination degree than Caohai Lake, and the effects of the landscape pattern indexes lagged.

**Supplementary Materials:** The following supporting information can be downloaded at: https://www.mdpi.com/article/10.3390/su15043145/s1, Figure S1: Land-use change pattern in Dianchi Lake Basin from 1990 to 2020; Table S1: Water-quality data of Waihai Lake from 1990 to 2020; Table S2: Water-quality data of Caohai Lake from 1990 to 2020; Table S3: Landscape metric data of the Dianchi Lake Basin from 1990 to 2020; Table S4: Classification of the CWQI values; Table S5: Standards used to evaluate the coupling coordination state for the water quality and land-use structure systems in Dianchi Lake Basin.

**Author Contributions:** Conceptualization, Z.Z. and X.L.; methodology, Z.Z. and X.L.; validation, X.L. and Z.H.; formal analysis, X.L. and Z.H.; writing—original draft preparation, Z.Z. and X.L.; writing—review and editing, J.L., W.Z. and H.G.; supervision, X.L.; funding acquisition, Z.Z. and X.L. All authors have read and agreed to the published version of the manuscript.

**Funding:** This research was funded by the National Natural Science Foundation of China, Grant numbers 42171240 and 31901322; by the Research Start-Up Fund from Southwest Forestry University, grant number 112125; by the Yunnan Dianchi Lake Protection and Management Foundation, Biodiversity Survey in Dianchi Lake Basin, grant number 2063010.

**Institutional Review Board Statement:** Not applicable.

**Informed Consent Statement:** Not applicable.

**Data Availability Statement:** Not applicable.

**Acknowledgments:** Our special thanks go to Yang Yuming and Pan Min for providing constructive suggestions.

**Conflicts of Interest:** The authors declare no conflict of interest.

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
