# Peer review of "Impact of Land Use/Land Cover and Landscape Pattern on Water Quality in Dianchi Lake Basin, Southwest of China"

_sustainability, doi:10.3390/su15043145_

Round 1

Author Response

Response to Reviewer 1 Comments

Comments and Suggestions for authors:

Review:

After reviewing the manuscript entitled "Impact of land-use/land-cover and landscape pattern on water quality in Dianchi Lake Basin, Southwest of China ". The authors have evaluated the effect and relative influence of LULC and landscape patterns on water quality in Dianchi Lake basin. I am convinced that this manuscript is a nice piece of work and can be considered for publication after improvement.

Abstract:

The abstract is well written, covering all necessary information.

Answer: Thank you for your positive comments.

Introduction:

I like this section. It's well written! The used references are more relevant and recent to the topic.

Answer: Thank you for your positive comments.

Materials and Methods:

Please add the explanation for subfigures (a, b and c) in figure 2.

Answer: Thanks for pointing that out. We have added explanations to the three sub-figures in Figure 2 as follows:

Figure 2. Geographical location of the study area

Note:The location of Dianchi Lake in China (a); Elevation of Dianchi Lake Basin (b); Location of 10 water quality monitoring sampling sites in Dianchi Lake (c).The map images we use come from Environmental Sciences, Chinese Academy of Sciences(http://www.resdc.cn/).

Regarding Table 2, It is not clear how the weights were assigned to landscape indices. What is the use of those weights in the study?

Answer: Thank you very much for pointing out the problem. In Table 2, the weights of landscape index and water quality index were objectively assigned by entropy method. The weights of these indicators are used in the TOPSIS comprehensive evaluation model. Generally speaking, Entropy - TOPSIS method is to first use entropy method to obtain newdata (original data multiplied by the weight calculated by entropy method), and then use the newdata for TOPSIS comprehensive analysis. The comprehensive situation of the landscape pattern and water quality of Dianchi Lake basin from 1990 to 2020 was evaluated year by year by using the Entropy-TOPSIS method in this paper. Then, the comprehensive evaluation results are used for coupled coordination analysis to explore the coordination level of landscape pattern and water quality. Now, we have added a statement about the principle of weights allocation and the use of those weights in the corresponding position of the paper.

Methodology for determining Land use change patterns is not described.

Answer: Thank you very much for pointing out the problem. We have added the following explanation in the article:

2.2.1 Land use data acquisition and processing

The remote sensing image data of Dianci Lake Basin every year from 1990 to 2020 was downloaded from the geospatial data cloud(http://www.gscloud.cn), and the land use information was extracted by using ENVI5.3 software based on the sample oriented object-oriented supervised classification method. According to the national classification standard of land use status (GB/T 210-2017) and combined with the actual situation of land use in Dianchi Lake Basin and the operability of interpretation, land use types were divided into eight categories: high forest land, shrub land and open forest land, grassland, farmland, water, wasteland and construction land. With the help of the actual land use data collected and recorded in the field, the accuracy was verified by the confusion matrix, and the Kappa coefficient was 0.91, which met the research accuracy requirements. The results of land use change are shown in Figure S1.

Results and Discussion:

Please provide results of Land use and land cover change patterns in supplementary material and describe their dynamics in the results sections.

Answer: Thanks for pointing that out. We have provided the results of the land use/land cover change patterns in the supplementary material as follows:

Figure S1. Land use change pattern in Dianchi Lake Basin from 1990 to 2020

We have described the dynamic changes of land use/land cover change patterns in the results sections as follows:

3.Results

3.1 Dynamic changes of land use in Dianchi Lake Basin

The area of farmland, high forest land, shrub land and open forest land decreased by 332.30km2, 70.85km2, 46.71km2 and 15.24km2, respectively. The area of built-land, grassland, bare land and water body increased by 392.13km2, 34.59km2, 30.43km2 and 9.74km2, respectively.Specially, the area of farmland was decreased from 26.90% in 1990 to 14.68% in 2020, and the built land was increased from 8.05% in 1990 to 21.98% in 2020. The major land use types in Dianchi Lake Basin are forest land, water, farmland in 1990, and the major land use type became built land in 2020.The spatial distribution of land use types (except Dianchi water) in Dianchi Basin showed that it became built land from farmland in the central Dianchi Lake Basin, and the forest land is mainly distributed in the northern and southern parts of the basin.

Regarding Table 5, Line no. 512 “The relative importance of various factors in affecting water quality and landscape indexes in Dianchi Basin”, and concluding remarks in Line no. 532-533 “The landscape pattern index of Waihai is more decisive for the coupling coordination degree than that of the Caohai Lake”. It is still not clear which factors affect coupling coordination between water quality and landscape indices. Are the factors different from indices? If so, please revise the results from Line no. 515 to 533. Otherwise, change the caption of table 5. It is confusing/misleading.

Answer: Thank you very much for pointing out the problem. We are very sorry for the confusion and misdirection caused by our unclear expression. We established a geographic detector model with coupling coordination degree as dependent variable and water quality and landscape pattern system index as independent variables. The objective is to identify the main indicators affecting the evolution of coupling coordination degree of Caohai Lake and Waihai Lake. So, the factors and indices are the same in this paper. We have corrected the caption of Table 5: " Table 5 Detection results of main indicators affecting coupling coordination degree variation",and the expressions and statements in the corresponding section are checked and corrected.

Conclusions:

Bullet (4) should be rewritten once results relevant to Table 5 are revised.

Answer: Thank you very much for pointing out the problem. We have rewritten this section as follows:

(4)There are different indexes affecting the coupling coordination degree of Caohai and Waihai Lake, but in general, landscape pattern is more decisive to the evolution of coupling coordination degree, and the landscape pattern indexes of Waihai Lake (mainly LPI and LSI) are more decisive to the coupling coordination degree than Caohai Lake.

Please also provide the limitations of the study.

 Answer: Thank you very much for pointing out the problem. We have rewritten this section as follows:

Although this paper explores the impact of land use and landscape pattern changes in the study area on the water quality of Dianchi Lake in the past 30 years, it can provide some references for the ecological protection and governance of the basin, but there are still limitations:(1) In different years and seasons, the complex relationship between landscape patterns and water quality parameters involves complex mechanisms and processes, which require further research.(2) This paper explores the process and relationship of historical change. The use of better models and methods as well as the simulation and prediction of future changes in the relationship between water quality and landscape under different development scenarios are worthy of further exploration.

References:

The references list needs a double check.

Answer:Thank you for pointing out this problem, we have checked and updated.

Reviewer 2 Report

Dear editor,

The paper is within the scope Journal, and it needs major revisions before it can be accepted as following:

1.      The abstract needs more developments and I suggest to be re-written.

2.      All the basic equations, therefore, an appropriate reference is necessary here.

3.      The results are needs to compare with other researchers.

4.      In the conclusion, please provide a clear justification for your work in this section.

5.      I do suggest to rewrite the discussion and results sections, better highlight the purpose of this study and the gap that the Authors would like to fill in the scientific literature, the practical aspects of such a study, and improve the readability of the paper.

6.      Authors should provide insights on recommendations for future work that might be used by others.

7.      Finally, Please, revise the manuscript by English native speaker because there are many grammatical errors within the manuscript,

With kind regards,

Author Response

Response to Reviewer 2 Comments

Dear editor,

The paper is within the scope Journal, and it needs major revisions before it can be accepted as following:

1. The abstract needs more developments and I suggest to be re-written.

Answer: Thank you very much for pointing this out. We have re-written the abstract.

Abstract:

The water quality of the basin is pronouncedly affected by surrounding land use types. Analyzing the impact of LULC and landscape patterns on water quality is critical for identifying potential drivers. In order to further study how LUCC affect water quality in typical plateau lake basin, this study investigated the impacts of land use types on water quality in Dianchi Lake Basin in Southwest China, we analyzed changes of land use types and landscape pattern of Dianchi basin,  calculated CWQI (Canadian Water Quality Index) value based the water quality indexes (PH, total phosphorus (TP), total nitrogen (TN), chemical oxygen demand (COD), dissolved oxygen (DO), permanganate index (CODMn), five-day biochemical ox-ygen demand (BOD5), ammonia nitrogen (NH3-N), Turbitidy, and chlorophyll-a (Chla)), and used RDA (Redundancy Analysis) , SMLR (Stepwise multiple linear regression) methods and coupling degree and coupling coordination degree, geographical detector model to explore the relation between water quality and land use type changes. The results show that: (1(1) Changes of the land use types were obvious, the major land became built land in 2020 from forest land, farmland in 1990(except Dianchi water).Landscape pattern indexes indicated that almost all of land use types were first scattered than gathered from1990 to 2020; (2) Changes of the water quality of Dianchi Lake was lagged the land use types changes, and the variation trends was similar to landscape pattern. The CWQI value was nearly linearly decreased from 1990 to 1998, it had a slight change in 1999-2013, and it was quickly increased from 2013; (3) Land use types had tight correlation with Dianchi water quality, and LPI was the most dominant factor in both Caohai and Waihai Lake. (4) There are different indexes affecting the coupling coordination degree of Caohai and Waihai Lake.

2.All the basic equations, therefore, an appropriate reference is necessary here.

Answer: Thank you very much for pointing this out. We have added appropriate references in where all the basic equations are located.

3. The results are needs to compare with other researchers.

Answer: Thank you very much for pointing this out. We have added this part.

4. In the conclusion, please provide a clear justification for your work in this section.

Answer: Thank you very much for pointing this out. We have re-written the conclusion.

Conclusion:

Based on LULC and water quality in Dianchi Basin from 1990 to 2020, we analyzed the relationship between landscape index and Dianchi Lake water quality, constructed a coupled model of landscape index and water quality system, analyzed the decisive influencing factors using geographic detectors, and conclude that:

(1) Changes of the land use types were obvious and nonlinear, the large farm land became built land in the study period.

(2) The water quality of Dianchi Lake has spatial differences in the concentrations of TN, TP and BOD5 in Caohai and Waihai Lake, and this led the variation of CWQI value in in Caohai and Waihai Lake. The Caohai Lake water area is smaller and the pollution is more serious than that of the Waihai Lake.

(3) Different land use types and landscape indicators have different impacts on the water quality of the Caohai Lake and the Waihai Lake. Caohai Lake are LPI, LSI and bare land, whereas Waihai are LPI, building land, LSI and bare land, though the largest explanatory variable was LPI in Caohai and Waihai Lake.

(4) The coupling coordination types of Caohai and Waihai Lake from 1990 to 2020 were divided into basic coordination and moderate quality coordination. There are different indexes affecting the coupling coordination degree of Caohai and Waihai Lake, the landscape pattern indexes of Waihai Lake (mainly LPI and LSI) are more decisive to the coupling coordination degree than Caohai Lake, and the effects of the landscape pattern indexes were lagged.

5. I do suggest to rewrite the discussion and results sections, better highlight the purpose of this study and the gap that the Authors would like to fill in the scientific literature, the practical aspects of such a study, and improve the readability of the paper.

Answer: Thank you very much for pointing this out. We have re-written the the discussion and results sections.We sincerely hope that the improvement is satisfactory.

6.Authors should provide insights on recommendations for future work that might be used by others.

Answer: Thank you very much for pointing this out. We have re-written this part.

7.Finally, Please, revise the manuscript by English native speaker because there are many grammatical errors within the manuscript.

Answer: Thank you very much for pointing this out. We have corrected the grammar.

Reviewer 3 Report

I read with great interest the manuscript submitted by Zhuoya Zhang et al. for consideration of sustainability. This manuscript presents a great idea regarding the Impact of land use/land cover and landscape pattern on water quality in Dianchi Lake Basin. Although the manuscript presents a good dataset and addresses relevant research questions, I consider that it can be accepted after making some modifications, I added all notes inside the Pdf file as sticky notes (yellow colour).

Author Response

Response to Reviewer 3 Comments

I read with great interest the manuscript submitted by Zhuoya Zhang et al. for consideration of sustainability. This manuscript presents a great idea regarding the Impact of land use/land cover and landscape pattern on water quality in Dianchi Lake Basin. Although the manuscript presents a good dataset and addresses relevant research questions, I consider that it can be accepted after making some modifications, I added all notes inside the Pdf file as sticky notes (yellow colour).

  1. Grammatical problems

Answer: Thank you for your careful modification. We have corrected all the grammar problems you pointed out.

  1. The link of “anthropogenic activities”

Answer: Thank you very much for pointing out the problem. We deleted the link.

  1. Mention some of the studies.

Answer: Thank you very much for pointing out the problem.We add some studies in the corresponding place.

  1. Figure2—It would be good to add some cities surrounding the study area and some land use patterns around the area

Answer: Thank you very much for your suggestion. We respond politely and respectfully. We decided not to add administrative units around the study area due to some requirements for map production and standardized use. The map of land use change in the study area from 1990 to 2020 can be seen in Figure S1.

Figure S1. Land use change pattern in Dianchi Lake Basin from 1990 to 2020

  1. This paragraph is repeated(Lines 122 -131)

Answer: Thank you very much for pointing out the problem.We have removed the duplicate paragraph.

  1. 1 Temporal and spatial variation of water quality in Dianchi Lake-it would be nice if a map added

Answer: Thanks for pointing that out. A map of each water quality index is indeed a good idea to express spatial differences, again thanks for your suggestion. We express our opinions politely and respectfully. Since Caohai Lake and Waihai Lake are two sub-lakes of Dianchi Lake, the two are separated by Caohai Tunnel (Caohai Lake is the northern part of Dianchi Lake and Waiha Lake is the southern area). We pay more attention to the north-south differences in the overall conditions of the two sub-lakes to provide different management and governance insights for the protection of the lakes. Therefore, we decided to use the average value of water quality monitoring stations located in the two sub-lakes to compare the North-South spatial differences in water quality, as shown in Figure 3.

  1. 3—Temporal variation in landscape patterns in Dianchi Lake Basin-Showing numbers and patterns in a map will be good to illustrate changes and stages of development and change

Answer: Thank you very much for pointing out the problem.We added the corresponding map, as shown in Figure S1.

  1. 6—Identification of the factors affecting the spatio-temporal evolution of the coupling coordination between landscape patterns and water quality in Dianchi Basin.-The authors may consider concise the flow of this subtitle for the ease of the readers, the paragraphs below need to be rewritten and simplified.

Answer: Thank you very much for pointing out the problem. We have simplified the subtitle and the paragraphs.

  1. Patents—It is preferable to delete this part, as there is no need for it. Numerous studies around the world have dealt with the same subject and with similar methodologies.

Answer: Thank you very much for pointing out the problem. We deleted this part.

References

Answer: Thank you very much for pointing out the problem. We double checked the references list.

Round 2
